# Comparison of the Effects of Recombinant and Native Prolactin on the Proliferation and Apoptosis of Goose Granulosa Cells

**DOI:** 10.3390/ijms242216376

**Published:** 2023-11-15

**Authors:** Donghang Deng, Wen Li, Xiaopeng Li, Xin Yuan, Liang Li, Jiwen Wang, Chunchun Han, Shenqiang Hu

**Affiliations:** 1State Key Laboratory of Swine and Poultry Breeding Industry, College of Animal Science and Technology, Sichuan Agricultural University, Chengdu 611130, China; dengdh94@163.com (D.D.); lixp0715@163.com (X.L.); cnliliang@foxmail.com (L.L.); wjw2886166@163.com (J.W.); 2Key Laboratory of Livestock and Poultry Multi-omics, Ministry of Agriculture and Rural Affairs, College of Animal Science and Technology, Sichuan Agricultural University, Chengdu 611130, China; 15520739895@163.com (W.L.); nihaoyuanxin88@outlook.com (X.Y.); 3Farm Animal Genetic Resources Exploration and Innovation Key Laboratory of Sichuan Province, Sichuan Agricultural University, Chengdu 611130, China

**Keywords:** goose, prolactin, protein purification, follicular development, proliferation, apoptosis

## Abstract

In poultry, prolactin (PRL) plays a key role in the regulation of incubation behavior, hormone secretion, and reproductive activities. However, previous in vitro studies have focused on the actions of PRL in ovarian follicles of poultry, relying on the use of exogenous or recombinant PRL, and the true role of PRL in regulating ovarian granulosa cell (GC) functions in poultry awaits a further investigation using endogenous native PRL. Therefore, in this study, we first isolated and purified recombinant goose PRL protein (rPRL) and native goose PRL protein (nPRL) using Ni-affinity chromatography and rabbit anti-rPRL antibodies-filled immunoaffinity chromatography, respectively. Then, we analyzed and compared the effects of rPRL and nPRL at different concentrations (0, 3, 30, or 300 ng/mL) on the proliferation and apoptosis of both GCs isolated from goose ovarian pre-hierarchical follicles (phGCs) and from hierarchical follicles (hGCs). Our results show that rPRL at lower concentrations increased the viability and proliferation of both phGCs and hGCs, while it exerted anti-apoptotic effects in phGCs by upregulating the expression of *Bcl-2*. On the other hand, nPRL increased the apoptosis of phGCs in a concentration-dependent manner by upregulating the expressions of *caspase-3* and *Fas* and downregulating the expressions of *Bcl-2* and *Becn-1*. In conclusion, this study not only obtained a highly pure nPRL for the first time, but also suggested a dual role of PRL in regulating the proliferation and apoptosis of goose GCs, depending on its concentration and the stage of follicle development. The data presented here can be helpful in purifying native proteins of poultry and enabling a better understanding of the roles of PRL during the ovarian follicle development in poultry.

## 1. Introduction

The orderly and progressive growth and development of ovarian follicles is the basis of egg production in poultry. These follicles, at different stages of development within the ovary of a goose during the egg-laying period, are generally categorized into the pre-hierarchical (<10 mm in diameter) and hierarchical (also called preovulatory, i.e., F6–F1) follicles [1]. Each follicle is mainly composed of three types of cells, including oocytes, granulosa cells (GCs), and theca cells. Among them, the GCs form the layers surrounding the oocytes, and are mainly responsible for synthesizing and secreting hormones and cytokines and providing nutrients for oocyte maturation and follicular development [2]. In addition, previous studies have indicated that the proliferation, differentiation, apoptosis, and autophagy of GCs play important roles in regulating ovarian follicular development, selection, and atresia [3,4].

Prolactin (PRL), known as a polypeptide hormone and mainly synthesized and secreted by lactotroph cells in the anterior pituitary gland, plays a crucial role in regulating female reproductive functions such as incubation behavior, hormone secretion, and follicular development in poultry [5,6]. It is widely accepted that the seasonal changes in breeding activity of both wild and domestic birds are attributed to the alternating secretions of luteinizing hormone (LH) and PRL, with plasma PRL levels reaching their peak, while LH levels decline sharply during the broodiness period [7]. Many studies have also demonstrated that the elevated level of plasma PRL is the main factor that induces an incubation behavior in poultry [8]. The influence of PRL on avian seasonal reproductive activities depends on its plasma levels, which are strongly regulated by a photoperiod [9]. It has been reported that, following immunization with recombinant PRL, a significant reduction in incubation behavior and altered egg production performance are observed in both turkeys and chickens [10,11]. Similarly, the number of large white follicles in a hen ovary was observed to be higher in PRL-immunized groups than in BSA-immunized ones, suggesting that PRL plays an important role in regulating the ovarian follicular development of poultry [11,12]. Abundant evidence supports the indirect and direct actions of PRL on the hypothalamic–hypophyseal–ovarian axis that regulates the incubation behavior and ovarian activities of poultry [6,9,13]. Furthermore, several recent in vitro studies using exogenous or recombinant PRL have demonstrated that PRL can act directly on both basal and gonadotrophin-mediated follicular cell steroidogeneses, depending on its concentration, the follicular size, and the stage of the ovulation cycle in chickens [14,15]. Additionally, lower concentrations of recombinant PRL promoted in vitro oocyte maturation and early embryonic development in mice [16], while the mice lacking *PRL* or *PRL* receptors exhibited complete infertility and abnormal ovarian follicular development [17,18]. Taken together, these results indicate that PRL has a dual role in either promoting or inhibiting ovarian follicular development, which is supposed to depend on its concentrations and the poultry species involved.

However, considering that previous studies on the actions and mechanisms of PRL in avian ovarian follicular cells rely on the use of exogenous or recombinant PRL rather than endogenous native PRL, deciphering its true roles awaits a further investigation. Moreover, compared to chickens and ducks, domestic geese generally show a stronger reproductive seasonality and, consequently, a lower egg production performance, which is assumed to be associated with the species/breed-specific pituitary secretions of PRL in response to photoperiodic changes [19,20,21,22,23,24]. Therefore, it is important to reveal the true role of PRL in regulating goose ovarian cell functions. In the present study, we firstly generated and purified the polyclonal antibodies against the recombinant goose PRL (rPRL). Subsequently, we constructed an immunoaffinity chromatography (IAC) column to isolate and purify the endogenous native PRL (nPRL) from geese pituitary glands. Finally, we compared the effects of rPRL and nPRL at different concentrations on the proliferation and apoptosis of the GCs isolated from goose ovarian pre-hierarchical follicles (phGCs) and hierarchical follicles (hGCs). These data will not only provide a better understanding of the role of PRL during avian ovarian follicular development, but also be helpful for controlling reproductive seasonality in practical goose production.

## 2. Results

### 2.1. Amplification of PRL Gene and Construction of Recombinant pET-28a-PRL

The genomic DNA extracted from the pituitary gland of Sichuan White Goose served as the template for PCR amplification. The cDNA encoding the mature peptide of goose PRL, with an approximate length of 600 bp, was successfully obtained using RT-PCR, and was further validated with sequencing (Figure 1A). The recombinant plasmid pET-28a-*PRL* was successfully synthesized by T4 ligase after cleavage of PRL and pET-28a fragments using restriction endonucleases Nco I and Xho I, and was validated by restriction enzymes digestion, as showed in Figure 1B.

### 2.2. Purification and Identification of Goose Recombinant PRL Protein and Its Antibody

The prokaryotic expression products of the pET-28a-*PRL* were obtained from the soluble fraction in *E. coli* BL21 (DE3) using different optimization conditions (Appendix A). The obtained supernatants were purified using Ni-affinity chromatography, and the results are shown in Figure 2A. The rPRL protein was collected after being washed with different concentrations of eluent buffer. The band of the recombinant protein appeared close to 25 kDa, which corresponds to the estimated molecular mass of the goose mature PRL. Then, the rPRL protein was characterized by quadrupole time-of-flight tandem mass spectrometry (Q-TOF-MS) and subsequent sequencing, and was verified to be the goose recombinant PRL protein (Figure 2B,C). These results demonstrate that the rPRL was successfully obtained in this study. In addition, the rabbit-anti-rPRL antibody, with a molecular mass of approximately 50 kDa, was purified using self-made rPRL protein-coupled affinity chromatography, as shown in Figure 2D.

### 2.3. Purification and Identification of Goose Native PRL Protein

The total protein extracted from the pituitary glands of Sichuan White Geese was lysed with tissue protein lysis buffer, followed by incubation on ice with an immunoaffinity chromatography column constructed using NHS-Activated beads and rabbit-anti-rPRL antibodies. These antibodies can specifically bind to the nPRL, allowing for a collection of native PRL proteins through the utilization of a low-pH eluent buffer (Figure 3A). Subsequently, the immunoreactivity of nPRL was tested using Western blot, with a rabbit-anti-rPRL antibody being the primary antibody (Figure 3B), resulting in two immunoreactive bands that appeared at approximately 23 kDa and 26 kDa (corresponding to the non-glycosylated and glycosylated PRL), respectively. These results demonstrate that the goose native PRL protein was successfully obtained in this study.

### 2.4. Effects of rPRL and nPRL on Goose Pre-Hierarchical and Hierarchical Granulosa Cell Viability

To investigate the effects of nPRL and rPRL on the viability of goose phGCs and hGCs, both cell types were treated with different concentrations of either nPRL or rPRL (0, 3, 30, or 300 ng/mL) for 24 h. The results show that both rPRL and nPRL at a concentration of 3 or 30 ng/mL significantly upregulated the viability of phGCs (*p* < 0.05), while 300 ng/mL of nPRL and rPRL showed no significant effects (Figure 4A). In hGCs, the nPRL exerted a stimulatory effect on cell viability in a concentration-dependent manner, whereas the trend for rPRL in hGCs was consistent with that in phGCs (Figure 4B).

### 2.5. Effects of rPRL and nPRL on Goose Pre-Hierarchical and Hierarchical Granulosa Cell Proliferation

To further investigate the effects of nPRL and rPRL on the proliferation of phGCs and hGCs, we conducted the EdU assay. As shown in Figure 5, our results show that treatment with 30 ng/mL of nPRL significantly promoted (*p* < 0.05), while nPRL at 3 or 300 ng/mL did not significantly affect (*p* > 0.05), the proliferation of phGCs. However, in hGCs, compared to the control group, neither rPRL nor nPRL treatments significantly changed the cell proliferation ratio (*p* > 0.05).

### 2.6. Effects of rPRL and nPRL on Goose Pre-Hierarchical and Hierarchical Granulosa Cell Apoptosis

We further examined the effects of rPRL and nPRL on the apoptosis of goose phGCs and hGCs using a flow cytometer. As shown in Figure 6, compared to the control group, 3 ng/mL of nPRL had no significant effect on the apoptotic rates of both phGCs and hGCs. However, as the concentration of nPRL increased, it significantly increased the apoptotic rates of both phGCs (300 ng/mL) and hGCs (30 and 300 ng/mL). In contrast, lower concentrations of rPRL (3 and 30 ng/mL) significantly reduced the apoptotic rate of phGCs (*p* < 0.05), while 300 ng/mL of rPRL showed no significant effect. In hGCs, the apoptotic rate was not significantly changed by rPRL at any concentration (*p* > 0.05).

### 2.7. Effects of rPRL and nPRL on the Expression of Several Key Genes Involved in Cell Apoptosis in Goose Pre-Hierarchical and Hierarchical Granulosa Cells

To further evaluate the effects of rPRL and nPRL on the apoptosis of goose phGCs and hGCs, the mRNA expression levels of *Bcl2*, *Caspase3*, *Fas,* and *Beclin1* were detected by qPCR. As depicted in Figure 7, treatment with rPRL at doses of 3 or 30 ng/mL significantly upregulated the mRNA expression of the anti-apoptotic marker gene *Bcl2* (*p <* 0.05). On the other hand, nPRL inhibited the expression of *Bcl2*. Meanwhile, nPRL promoted the expression of *Caspase3* and *Fas* in phGCs in a dose-dependent manner. Both rPRL and nPRL at medium to high concentrations significantly reduced the mRNA levels of *Beclin1* in phGCs and hGCs (*p <* 0.05). However, neither rPRL nor nPRL significantly changed the expressions of *Bcl2*, *Caspase3*, and *Fas* in hGCs.

## 3. Discussion

Prolactin is widely recognized as one of the main factors causing incubation behavior, implying a negative role in the regulation of poultry ovarian follicular development and egg production performance [25]. However, several recent studies in chickens have reported that PRL may play a stimulatory role in ovarian follicle development by influencing the synthesis and secretion of follicular cell steroid hormones, depending on its concentration and the stage of follicle development [14,15]. Considering that almost all previous in vitro studies on the roles of poultry PRL used either exogenous PRL purified from other species [26] or recombinant PRL [27], deciphering its exact roles awaits a further investigation using endogenous native PRL. Thus, herein we firstly isolated and purified the nPRL, and subsequently compared the effects of rPRL and nPRL on the proliferation and apoptosis of goose phGCs and hGCs.

The prokaryotic expression vector pET-28a-PRL was constructed to express the rPRL protein in *E. coli*. The rPRL was purified using Ni-affinity chromatography, and was then used to immunize rabbits for preparing the rabbit-anti-rPRL polyclonal antibodies. Compared to our previous research [28], in this study, we replaced the prokaryotic expression vector pET-32a with pET-28a to optimize the recombinant plasmid construction efficiency and to reduce the molecular mass of the rPRL protein from 37 kDa to 25 kDa, which may be closer to that of the nPRL protein. Then, the nPRL protein was purified from the pituitary glands of Sichuan White Geese using an immune-affinity column. Thus, this study represents the first to obtain a highly pure nPRL and to investigate its effects on follicular cell functions.

Previous studies have suggested that PRL can act directly on GCs to regulate avian ovarian follicular development [29,30]. In this study, both goose phGCs and hGCs were treated with our purified rPRL and nPRL at different concentrations, respectively. Our results show that both rPRL and nPRL, at a low concentration, increased the viability of phGCs and hGCs, but only nPRL at moderate-to-high concentrations promoted the viability and proliferation of phGCs. Meanwhile, we also found that nPRL at moderate-to-high concentrations exerted pro-apoptotic effects on both phGCs and hGCs, while lower concentrations of rPRL decreased the apoptotic rate of phGCs. These data collectively demonstrate the differences in the bioactivity of rPRL and nPRL in goose ovarian GCs, and such differences could be due to the fact that the rPRL is an unfolded and post-translationally unmodified protein, while the nPRL is folded and post-translationally modified. In support of this, it has been previously suggested that the folded chicken PRL showed high bioactivities in a homologous system [31]. Moreover, there is evidence that post-translational modifications of chicken PRL significantly affected both its biological activities during the reproductive cycle [32] and ovarian follicular cell steroidogenesis [15]. Regarding the effects of PRL on avian ovarian follicular cell functions, it has been previously shown that PRL regulated both the basal and gonadotropin-stimulated estrogen and progesterone production in chicken ovarian follicular cells, depending on the concentration, the degree of glycosylation, the type of gonadotropin, and the stage of follicle development [14,15]. Besides this, it is noticeable that this study represents the first to reveal the effects of PRL on the viability, proliferation, and apoptosis of avian GCs. Consistent with our results, a low concentration of PRL significantly increased the viability and antioxidant capacity, while high PRL concentrations induced the oxidative stress and apoptosis of ovine ovarian GCs [33,34]. A recent study reported that plasma PRL levels, ranging from 25 to 100 ng/mL, could be beneficial for metabolic fitness, which is crucial for maintaining metabolic homeostasis [35]. The circulating PRL, at either extremely low or high levels, disrupted the metabolic homeostasis in diabetes and induced obesity-derived metabolic diseases [36,37]. It has also been reported that moderate levels of PRL played a positive role in glucose-stimulated insulin secretion, while higher PRL levels impaired the insulin-secretory capacity in diabetic mice [38]. Similar to our results, PRL exerted a biphasic effect on the growth of human endometrial cells, playing a positive role at low concentrations and a negative role at high concentrations [39]. Thus, it was proposed that a FIT-PRL concentration ranging from 3 to 30 ng/mL could promote the viability and proliferation, while high PRL concentrations induce the apoptosis, of goose GCs, depending on the stage of follicle development. 

It is well known that apoptosis is a form of programmed cell death, characterized by a series of morphological and biochemical changes, which is attributed to accurately regulated molecular events or signaling cascades. The Bcl-2 family plays an important role in the regulation of the mitochondria-mediated pathway of apoptosis. Among them, the anti-apoptotic protein Bcl2 is located on the surface of the mitochondrion and inhibits the progression of apoptosis by regulating the permeability of the mitochondrial outer membrane [40]. As a member of the tumor necrosis factor receptor superfamily, the Fas-mediated oligomerization and conformational changes via ligand binding can result in downstream caspase activation and cell apoptosis [41]. Moreover, caspases are the conserved executioners of apoptosis, and can be activated by the upstream apoptotic signals emanating from external and internal stimuli [42]. In the present study, our results show that low concentrations of rPRL significantly upregulated the levels of *Bcl-2* to inhibit the apoptosis of phGCs, which is consistent with the observations made using flow cytometry. However, the expression of *Bcl-2* was downregulated after treatment with nPRL, and higher concentrations of nPRL increased the apoptotic rate of hGCs by upregulating the expressions of *Fas* and *Caspase3*. Similar to what was found in our study, with increasing concentrations of ovine PRL, the viability of ovine ovarian GCs increased first and then decreased. Moreover, a low concentration (below 20 ng/mL) of PRL decreased the apoptosis of GCs, while 500 ng/mL of PRL increased the apoptosis and inhibited the steroid hormone secretion of GCs [43]. These results suggest that the effects of PRL on the proliferation and apoptosis of either mammalian or avian GCs depend on its concentration. Becn1, known as an autophagy-related protein, plays a role in preventing the assembly of the pre-autophagosomal structure and inhibiting autophagy by binding to Bcl-2 [44]. It has been reported that autophagy is involved in regulating the apoptosis of GCs to accelerate follicular atresia [45]. In this study, the expression levels of *Becn1* and *Bcl-2* in phGCs and hGCs were significantly decreased after treatment with high concentrations of either rPRL or nPRL. There is evidence that Becn1 plays important roles in the regulation of the life span of human corpus luteum and ovarian androgen-secreting cells by maintaining autophagy at levels that promote cell survival rather than cell death [46]. Additionally, the siRNA-mediated knockdown of *Becn1* showed that the inhibition of autophagy resulted in a decreased expression of genes involved in the differentiation of GCs [47]. These results imply that high PRL concentrations may promote the apoptosis of goose GCs, at least by part, by disrupting the autophagy process. 

## 4. Materials and Methods

### 4.1. Ethics Statement

All experimental procedures involving the manipulation of animals in this study were conducted in concordance with the “Guidelines for Experimental Animals” of the Ministry of Science and Technology (Beijing, China). This study has been reviewed and approved by the Sichuan Agricultural University Animal Ethical and Welfare Committee (Approval No.: 20190035).

### 4.2. Birds and Tissue Collection

All the experimental geese were provided by the Waterfowl Breeding Experimental Farm of Sichuan Agricultural University (Ya’an Campus, Sichuan, China). A total of 50 female Sichuan White Geese were used for tissue collection. Thirty Sichuan White Geese were euthanized by carbon dioxide, and the pituitary glands were immediately isolated. One of the pituitary glands was used for amplifying the cDNA encoding the mature peptide of PRL, and the others were used for the purification of nPRL protein. Twenty Sichuan White Geese, aged between 35 and 45 weeks and selected based on their egg-laying records to ensure regular sequences of at least 2–3 eggs, were used for the isolation of phGCs and hGCs.

### 4.3. Construction of the Goose Recombinant pET28a-PRL

Genomic DNA was extracted from the pituitary gland of a Sichuan White Goose using the FastPure Cell/Tissue DNA Isolation Mini Kit (Vazyme, Nanjing, China). According to the reported nucleotide sequence of the Sichuan White Goose *PRL* gene (GenBank accession number: GQ202542.1), the cDNA encoding the goose PRL mature peptide was obtained by polymerase chain reaction (PCR) amplification using our designed primers F (5′-CATGCCATGGGCTTGCCTATCTGCCCCAATGGATCTG-3′) and R (5′-CCCTCGAGGCAATTGCTATCATGTATTAGGCGGC-3′). The PCR was performed using a T100TM thermal cycler (Bio-Rad, Hercules, CA, USA) and the reaction conditions are as follows: 95 °C for 5 min; 30 cycles of 95 °C for 30 s, 65 °C for 30 s, and 72 °C for 45 s; 72 °C for 7 min. The PCR product was identified by 1.5% agarose gel electrophoresis and recovered by gel extraction kit (Omega Bio-tek, Norcross, GA, USA).

The purified PCR products were transformed into competent *E. coli* DH5á cells. Positive clones were screened by RT-PCR and sent to Tsingke Biotechnology Co., Ltd. (Chengdu, China) for sequencing analysis. Subsequently, both the PRL fragment and the pET-28a plasmid (Yeasen, Shanghai, China) were digested with NcoI and XhoI restriction enzymes, followed by ligation using the T4 DNA ligase at 16 °C for 12 h. Finally, the recombinant plasmid pET28a-*PRL* was introduced into *E. coli* BL21 (DE3) for protein expression.

### 4.4. Prokaryotic Expression and Purification of the Goose Recombinant PRL (rPRL) Protein

The *E. coli* BL21 (DE3) strain carrying the Pet28a-PRL plasmid was cultured in Luria–Bertani medium, supplemented with 50 g/L kanamycin at 37 °C, until reaching an OD600 of 0.4–0.6. Subsequently, the conditions for protein expression, including the concentration of isopropyl-β-D-thiogalactopyranoside (IPTG), induction temperature, and induction time, were assessed and optimized. Finally, protein expression was induced at 37 °C using a concentration of 0.5 mmol/L IPTG for a duration of 6 h. The culture medium was centrifuged at 10,000× *g* for 10 min to collect the sediment, which was broken by sonication for 30 min followed by centrifugation to remove the supernatants. The rPRL protein was finally purified using Ni affinity chromatography and analyzed by both sodium dodecyl sulfate–polyacrylamide gel electrophoresis (SDS-PAGE) and quadrupole time-of-flight tandem mass spectrometry (Q-TOF-MS).

### 4.5. Preparation and Purification of Polyclonal Antibodies against the rPRL Protein

The rabbits were immunized with our purified rPRL protein four times with a two-week interval between each immunization. Subsequently, the antiserum was collected from these rabbits. To purify the anti-rPRL antibodies from the antiserum, an empty column was filled using the CNBR-activated Sepharose 4B filler. Thereafter, 1 mg of rPRL protein together with the coupling solution (0.1 M NaHCO_3_, 0.5 M NaCl, pH 8.3) was introduced into the column at 4 °C for 12 h. Next, the sealing solution (Tris-HCl, pH 8.0) was added to fully seal any unbound protein sites for 6 h. The affinity chromatography column was prepared and cleaned with the appropriate phosphate-buffered saline (PBS), consisting of 50 mM NaH_2_PO_4_ and15 mM NaCl. The diluted serum was loaded onto the self-made columns and incubated at 4 °C for 12 h. After being washed three times with PBS, the antibodies were finally eluted using an elution buffer (0.15 mM NaCl, pH 2.5) at a flow rate of about 1 mL/min, and immediately frozen at −80 °C.

### 4.6. Preparation of the Immunoaffinity Chromatography (IAC) Column and Purification of the Goose Native PRL (nPRL) Protein

To purify the nPRL protein, the rabbit-anti-rPRL antibody was used for specific coupling with nPRL. In detail, the rabbit-anti-rPRL antibody was firstly coupled to the NHS-activated beads 4FF (Smart-Lifesciences, Changzhou, China) in a coupling buffer (0.2 M NaHCO_3_, 0.5 M NaCl, pH 8.0) at 4 °C for 1 h. Subsequently, the liquid in the column was discarded, and the column was sealed with a sealing solution (0.1 M Tris-base, pH 8.5) at 28 °C for 1 h. After being washed with PBS, the IAC column was successfully prepared. Then, the goose pituitary glands were lysed in a lysis buffer (Beyotime Biotech, Nantong, China) and centrifuged at 10,000× *g* at 4 °C for 10 min, and the resulting supernatant was loaded onto the self-made IAC column and incubated at 4 °C for 1 h. After being washed with PBS, the nPRL protein was finally eluted using an elution buffer (0.1 M Glycine, pH 3.0) at a flow rate of about 1 mL/min. The protein concentration measurement, SDS-PAGE, and Western blot analyses were performed to characterize the purified nPRL.

### 4.7. Cell Culture and Treatment

The ovarian follicles were obtained from the abdominal cavity of Sichuan White Geese during the egg-laying period, and the GC layers were subsequently isolated from both the pre-hierarchical (6–10 mm in diameter) and hierarchical follicles (F3-F1). After being washed with PBS, the isolated GC layers were cut into small pieces and subjected to digestion with 0.1% collagenase II. The resulting cell suspensions were then mixed with DMEM/F12 supplemented with 10% fetal bovine serum (Sigma, St. Louis, MO, USA) to halt the digestion process. After passing through a 200-mesh sieve and centrifugation at 1200 g/min for 10 min, the supernatant was discarded while the cells were resuspended in the aforementioned medium. The number of GCs in each follicular category was determined using a hemocytometer. Then, the cells were diluted to approximately 6 × 10^5^ cells /mL in media, seeded into different culture plates, and incubated at 37 °C under a humidified atmosphere containing 95% air and 5% CO_2_ until reaching a confluence of >70%. Following an initial medium change after 6 h of incubation, non-adherent cells were aspirated, while the remaining were used for subsequent treatments. After incubating for 24 h, both phGCs and hGCs were treated with nPRL and rPRL at the concentrations of 0, 3, 30, or 300 ng/mL for another 24 h, and were finally used for cell viability, proliferation, and apoptosis, as well as gene expression analysis. The concentrations of rPRL and nPRL were selected based on both the observed fluctuations in circulating PRL levels of female geese throughout the reproductive cycle [22,23] and the doses used in previous in vitro studies [14,15,31]. 

### 4.8. Cell Viability Assay

The viability of GCs was evaluated using the CCK-8 assay. Briefly, the GCs were firstly seeded into 96-well plates and cultured for 24 h. Then, the cells were treated with nPRL and rPRL at different concentrations (0, 3, 30, and 300 ng/mL) for another 24 h. After treatment, the CCK-8 solution (APExBIO, Houston, TX, USA) (10 µL/well) was added to each well and incubated at 37 °C for 4 h. Finally, the absorbance was measured at 450 nm using spectrophotometry.

### 4.9. EdU and Hoechst 33,342 Staining Assay

The GCs were firstly seeded onto 96-well plates and incubated for 24 h. After treatment with nPRL and rPRL at different concentrations for 24 h, the GCs were exposed to 50 μM of 5-ethynyl-2′-deoxyuridine (RiboBio, Nantong, China) at 37 °C for 4 h according to the manufacturer’s instructions. Then, the GCs were fixed with a solution containing 4% paraformaldehyde for 30 min and permeabilized with a solution containing 0.5% TritonX-100 for 10 min. After being washed with PBS, the GCs were incubated with a reaction cocktail consisting of Apollo dye (1×) in a volume of 100 μL for 30 min. Finally, the DNA content within cells was stained using Hoechst33342 dye for 30 min and visualized under a fluorescence microscope.

### 4.10. Annexin V-FITC/PI Double Staining in the Detection of Apoptosis by Flow Cytometry

The GCs were firstly seeded onto 24-well plates and incubated for 24 h. Subsequently, the GCs were treated with rPRL and nPRL at different concentrations for an additional 24 h. To assess the apoptosis rate induced by PRL, the cells were harvested and double-stained using the Annexin V-FITC/PI apoptosis detection kit (Vazyme Biotech, Nanjing, China) according to the manufacturer’s instructions. The apoptotic rate was detected using a BD Accuri C6 Flow cytometer and analyzed using FlowJo software (v10.8). A total of 10,000 cells per sample were analyzed, and the apoptotic rate was determined by summing up the cell proportions of early and late apoptotic cells.

### 4.11. Quantitative Real-Time PCR

The extracted RNA was reverse transcribed into cDNA using the Prime Script RT reagent kit (Takara, Dalian, China) according to the manufacture’s guidelines. The qRT-PCR reaction solution was prepared in a total volume of 12.5 μL containing 1 μL cDNA, 6.25 mL of 2× SYBR Premix Ex Taq (Vazyme Biotech, Nanjing, China), 4.25 μL of ddH2O, and 0.5 μL of each specific primer pair (10 μM). The reactions were conducted under the following conditions: pre-denaturation at 95 °C for 10 s, followed by 40 cycles of denaturation at 95 °C for 5 s and annealing/extension at the corresponding temperature of each primer pair for 30 s. An 80-cycle melting curve was performed, with the temperature ranging from 60 °C to 95 °C and increasing by 0.5 °C every 10 s. Each sample were amplified in triplicate, and the relative mRNA expression levels of target genes were normalized to the reference genes *GAPDH* and *β-Actin* using the comparative Cq method (ΔΔCq) [48]. The primer pairs used for qRT-PCR are listed in Table 1.

### 4.12. Statistical Analysis

The SPSS 22.0 and GraphPad Prism 8 software were used for data analysis. The results are presented as the mean ± S.D. Statistical significance was determined using one-way ANOVA followed by the Duncan’s multiple range test. Differences between groups were considered statistically significant at *p* < 0.05 and extremely significant at *p* < 0.01.

## 5. Conclusions

In conclusion, the present study obtained a highly pure nPRL for the first time. Moreover, our results have demonstrated that the rPRL at lower concentrations increased the viability and proliferation of both phGCs and hGCs, while exerting anti-apoptotic effects only in phGCs. By comparison, the nPRL showed more pronounced stimulatory effects on the viability and proliferation of both GC types, but increased their apoptosis at higher concentrations. These results suggest a dual role of PRL in regulating the proliferation and apoptosis of goose GCs, depending on its concentration and the stage of follicle development. These data presented here provide a better understanding of the roles of PRL in regulating avian ovarian follicular development, and can be helpful for purifying native proteins of poultry and for controlling reproductive seasonality in practical goose production. 

## Figures and Tables

**Figure 1 ijms-24-16376-f001:**
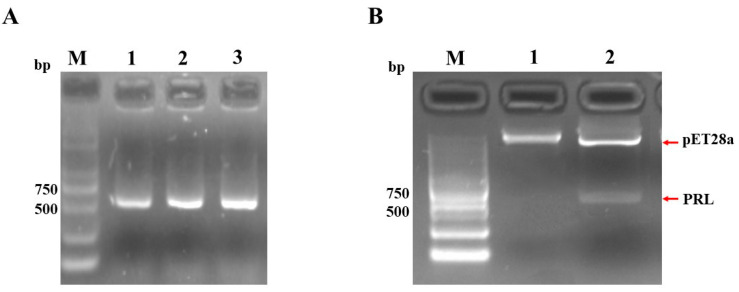
Amplification of PRL gene and construction of the recombinant goose pET-28a-PRL. (**A**) The PCR products of the cDNAs encoding goose mature PRL were visualized on an agarose gel. Lanes 1–3, the fragments of the PCR products of PRL. (**B**) Double enzyme digestion verification of the recombinant plasmid pET-28a-*PRL* plasmid. Lane 1: recombinant plasmid pET-28a-*PRL*. Lane 2, the double digestion products of recombinant plasmid pET-28a-*PRL* by digestion with the Nco I and Xho I restriction enzymes. M, DNA marker.

**Figure 2 ijms-24-16376-f002:**
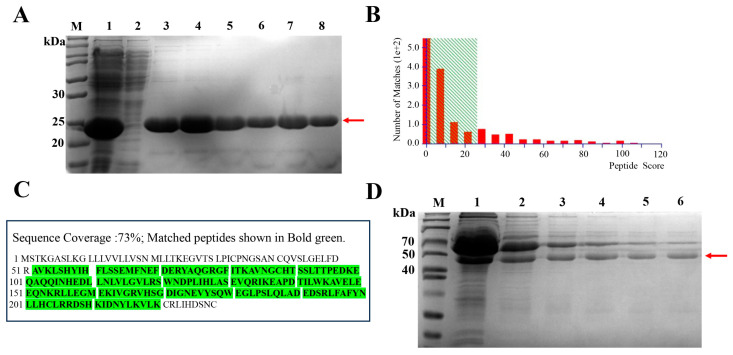
Purification and identification of goose recombinant PRL protein (rPRL) and the rabbit-anti-rPRL antibodies. (**A**) The SDS-PAGE results showing the rPRL solutions purified by Ni affinity chromatography. M, protein marker; Lane 1, the supernatant after lysis of pET-28a-PRL-BL21; Lane 2, the sample of the supernatant after flowing through the Ni column; Lane 3–8, the sample after cleaning the Ni column with eluent buffer. The red arrow refers to the band of rPRL. (**B**) Peptide score distribution of the recombinant protein analyzed by Q-TOF-MS. (**C**) Sequence coverage of the peptides characterized by Q-TOF-MS. These matched peptides are shown in bold green when compared with the goose mature PRL. (**D**) SDS-PAGE results of the rabbit-anti-rPRL sera before and after being purified using the rPRL protein-coupled affinity chromatography. Lane 1, the diluted rabbit-anti-PRL sera; Lanes 2–6, the solutions purified from the diluted rabbit-anti-rPRL sera using the Tris-HCl elution buffer. The red arrow refers to the band of the rabbit-anti-rPRL antibody.

**Figure 3 ijms-24-16376-f003:**
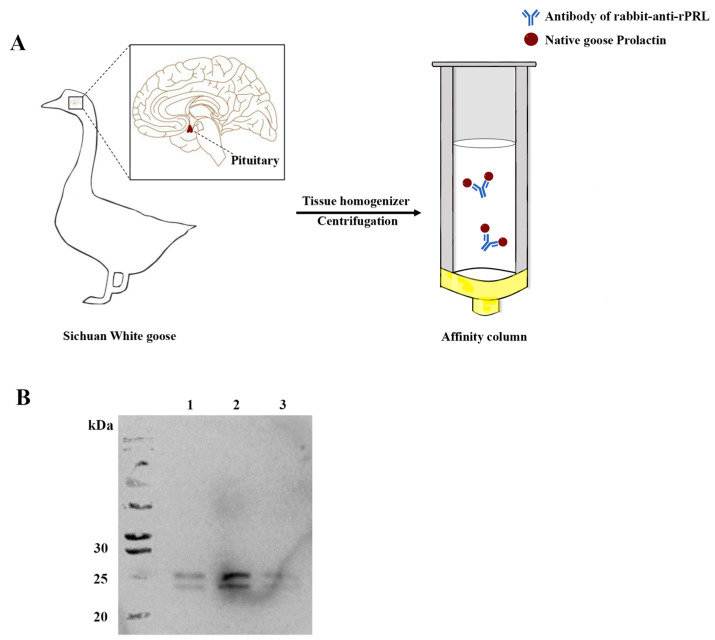
Purification and identification of the native PRL protein from Sichuan White Goose. (**A**) The steps of purification. (**B**) The Western blot results of the goose native PRL protein that was purified using a rabbit anti-rPRL antibodies-filled affinity chromatograph column. Lanes 1–3, the goose native PRL protein solutions that were eluted using 0.1 M Glycine at a pH of 3.

**Figure 4 ijms-24-16376-f004:**
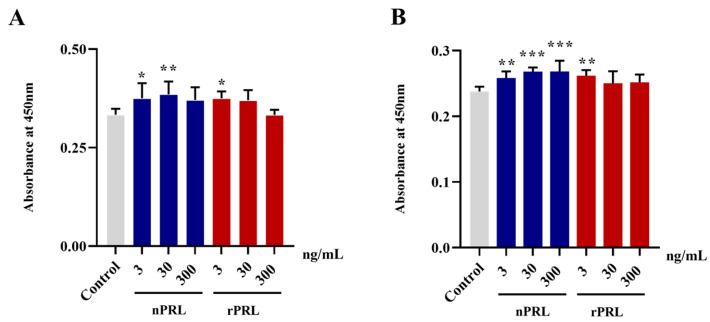
Comparison of the effects of rPRL and nPRL, at different concentrations, on the viability of goose phGCs (**A**) and hGCs (**B**). The viability of GCs was determined using the CCK-8 assay. Data are presented as the mean ± S.D. “***” indicates an extremely significant difference compared to the control group (*p* < 0.001), “**” indicates an extremely significant difference compared to the control group (*p* < 0.01), “*” indicates a significant difference compared to the control group (*p* < 0.05).

**Figure 5 ijms-24-16376-f005:**
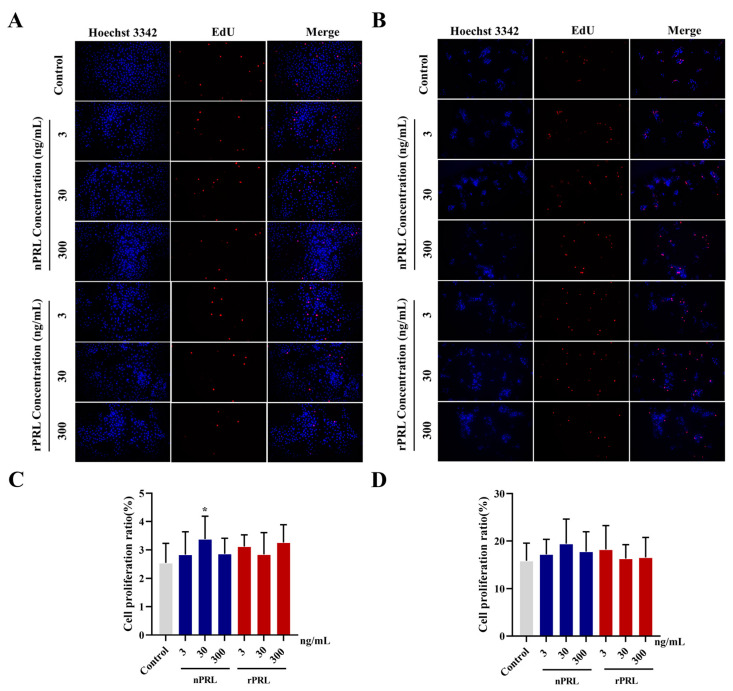
Comparison of the effects of rPRL and nPRL, at different concentrations, on the proliferation of goose phGCs (**A**,**C**) and hGCs (**B**,**D**). The cell proliferation was tested using the EdU assay. Data are presented as the mean ± SD. “*” indicated a significant difference compared to the control group (*p* < 0.05).

**Figure 6 ijms-24-16376-f006:**
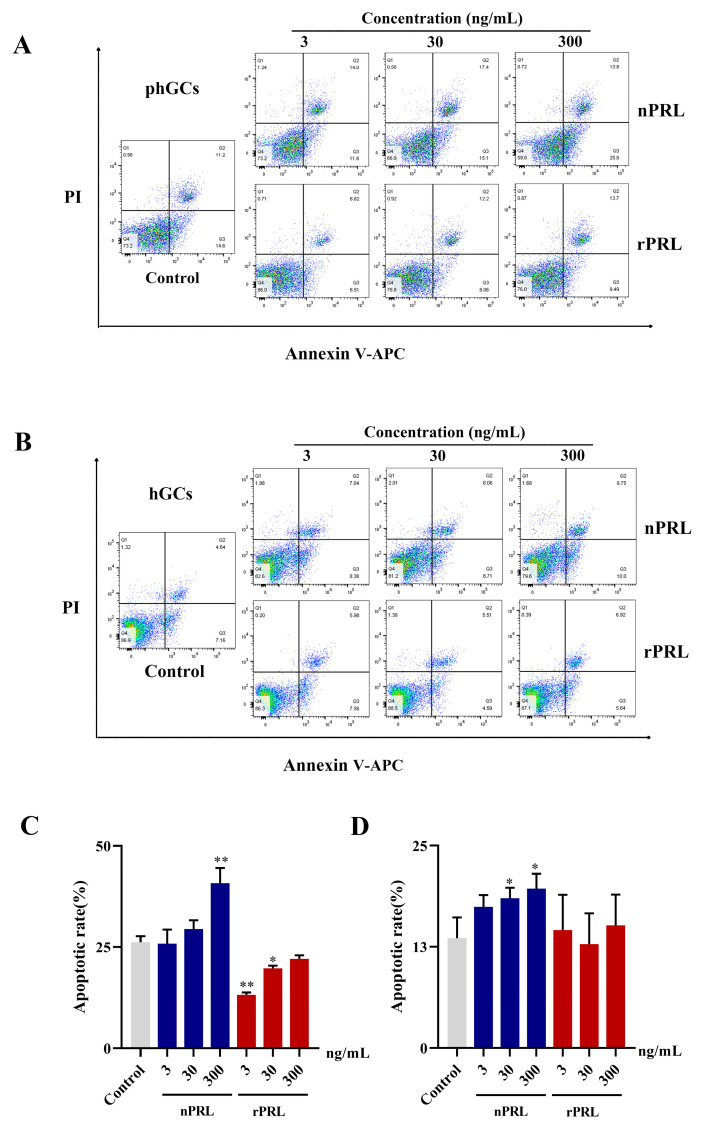
Comparison of the effects of rPRL and nPRL, at different concentrations, on the apoptosis of goose phGCs (**A**,**C**) and hGCs (**B**,**D**). The cells stained with annexin-V-APC were considered as apoptotic. The Q4 quadrant (Annexin V−/PI−), Q3 quadrant (Annexin V+/PI−), and Q2 quadrant (Annexin V+/PI+) represent the percentages of healthy cells, cells during early apoptosis, and cells during late apoptosis, respectively. The apoptotic rate was defined as the percentage of apoptotic cells: the cells during early and late apoptosis/total cells × 100%. Data are presented as the mean ± S.D. “**” indicates an extremely significant difference compared to the control group (*p* < 0.01), while “*” indicates a significant difference compared to the control group (*p* < 0.05).

**Figure 7 ijms-24-16376-f007:**
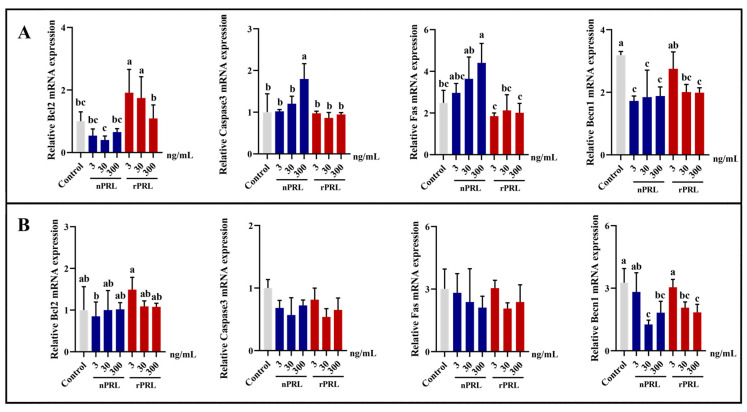
Effects of rPRL and nPRL on the expressions of several key genes involved in cell apoptosis in goose phGCs (**A**) and hGCs (**B**). Data are presented as the mean ± S.D. Different lowercase letters indicate significant differences among different treatment groups at *p* < 0.05.

**Table 1 ijms-24-16376-t001:** Primer pairs for real-time quantitative PCR.

Gene	Forward Primer (5′–3′)	Reversed Primer (5′–3′)	Tm (°C)	Product Length (bp)
Bcl2	CCTTCGTGGAGTTGTATGGCA	CCACCAGAACCAAACTCAGGATA	60	100
Caspase3	CTGGTATTGAGGCAGACAGTGG	CAGCACCCTACACAGAGACTGAA	60	158
Fas	CACTCCCACAAGTCAAG	AGTAGGGTTCCATAGGC	60	163
Becn1	CGCTGTGCCAGATGTGGAAGG	CAGAAGGAATACTGCGAGTTCAAGA	60	151
GAPDH	GCTGATGCTCCCATGTTCGTGAT	GTGGTGCAAGAGGCATTGCTGAC	60	86
β-Actin	CAACGAGCGGTTCAGGTGT	TGGAGTTGAAGGTGGTCTCG	60	92

## Data Availability

All data supporting our reported results have been given in the Results section of this article.

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
