# Peer review of "Comparison of the Effects of Recombinant and Native Prolactin on the Proliferation and Apoptosis of Goose Granulosa Cells"

_ijms, 2023, doi:10.3390/ijms242216376_

Round 1

Reviewer 1 Report

Comments and Suggestions for Authors

1.      There is no appropriate justification for undertaking the presented research, as well as no explanation of the purpose of the comparison of recombinant and native prolactin.

2.      The paper lacks an explanation of why such particular doses of PRL were selected for the experiment.

3.      The discussion is conducted inappropriately - there is no discussion of the obtained results (the effect of PRL on viability, proliferation and expression of genes related to apoptosis) with the results obtained in other bird species, with an indication of whether the effects were determined with the use of recombinant or native prolactin.

4.      Lack of summary of the effects of both rPRL and nPRL, as well as a lack of conclusions considering their similar/different effects of action.

5.      Determining the protein abundance of Bcl2, Caspase3, Fas and Beclin1 with the use of the Western-blot method would be a great addition to the research.

6.      Authors should pay more attention to abbreviations: 1/ in some places, introducing an abbreviation is unnecessary - it is not used in the text; 2/ some abbreviations are not explained; 3/ do not start a sentence with an abbreviation (e.g. the first sentence in the discussion section).

7.      In the results section, authors should avoid phrases such as: ‘These results suggested (…)’. They do not sound good in the context of the reliability and correctness of the obtained results.

8.      An inaccurate statement should be modified: lines 157-160 ‘(…) stimulatory effect was blocked as the concentration increases.' – the effect was not blocked, PRL at higher doses simply did not affect cell proliferation.

9.      Line 388 – lack of the word 'FITC'.

Comments on the Quality of English Language

The language needs to be checked and revised thoroughly.

Author Response

Response to Reviewer 1 Comments

  1. Summary

We thank you for your careful review of our paper (Manuscript Number: IJMS-2683907) entitled “Comparison of the effects of recombinant and native prolactin on the proliferation and apoptosis of goose granulosa cells”. According to your comments, we have made corresponding responses on a ‘point by point’ basis that are listed as follows, and the detailed revisions have also been red highlighted in this revised manuscript. We hope that this revised manuscript will now meet the publication standard.

Thank you for your careful review and valuable comments again!

Sincerely,

Donghang Deng

2.  Questions for General Evaluation

Reviewer’s Evaluation

Does the introduction provide sufficient background and include all relevant references?

Must be improved

Are all the cited references relevant to the research?

Yes

Is the research design appropriate?

Yes

Are the methods adequately described?

Yes

Are the results clearly presented?

Yes

Are the conclusions supported by the results?

Must be improved

  1. Response to Comments and Suggestions

Comments 1: There is no appropriate justification for undertaking the presented research, as well as no explanation of the purpose of the comparison of recombinant and native prolactin.

Response 1: Thank you for your valuable comments. As you pointed out, we have supplemented such information in the last paragraph of the Introduction section. Please see the detailed changes in our revised manuscript (Page 2, Lines 74-81).

Comments 2: The paper lacks an explanation of why such particular doses of PRL were selected for the experiment.

Response 2: As you pointed out, we have supplemented such information in the ‘cell culture and treatment’ part of the Materials and Methods section. Please see the detailed changes in our revised manuscript (Page 11, Lines 392-395).

Comments 3: The discussion is conducted inappropriately - there is no discussion of the obtained results (the effect of PRL on viability, proliferation and expression of genes related to apoptosis) with the results obtained in other bird species, with an indication of whether the effects were determined with the use of recombinant or native prolactin.

Response 3: As you pointed out, we have improved the discussion section by comparing our results with those obtained in other bird species and analyzing the divergent effects of goose recombinant and native prolactin. Please see the detailed changes in our revised manuscript (Page 8, Lines 238-254).

Comments 4: Lack of summary of the effects of both rPRL and nPRL, as well as a lack of conclusions considering their similar/different effects of action.

Response 4: As you pointed out, we have supplemented such information in Pages 8-9, Lines 233-238 and Pages 13, Lines 445-449 of our revised manuscript.

Comments 5: Determining the protein abundance of Bcl2, Caspase3, Fas and Beclin1 with the use of the Western-blot method would be a great addition to the research.

Response 5: We agree that determining the protein abundance of Bcl2, Caspase3, Fas, and Beclin1 would be a great addition to this study. However, lack of specific antibodies against goose Bcl2, Caspase3, Fas, and Beclin1 makes this investigation impossible at the present stage. We also tried using antibodies from other species, but the results were disappointing. Now we are cooperating with other labs to produce the goose specific antibodies for further investigations in the near future.

Comments 6: Authors should pay more attention to abbreviations: 1/ in some places, introducing an abbreviation is unnecessary - it is not used in the text; 2/ some abbreviations are not explained; 3/ do not start a sentence with an abbreviation (e.g., the first sentence in the discussion section).

Response 6: As you pointed out, we have carefully checked the usage of abbreviations and made corresponding revisions throughout our manuscript. Please see the detailed changes in our revised manuscript.

Comments 7: In the results section, authors should avoid phrases such as: ‘These results suggested (…)’. They do not sound good in the context of the reliability and correctness of the obtained results.

Response 7: As you pointed out, we have carefully checked the Results section and avoided the use of such phrases to reflect the reliability and correctness of the obtained results. Please see the detailed changes in our revised manuscript.

Comments 8: An inaccurate statement should be modified: lines 157-160 ‘(…) stimulatory effect was blocked as the concentration increases.' – the effect was not blocked, PRL at higher doses simply did not affect cell proliferation.

Response 8: As you pointed out, we have corrected this statement. Please see the detailed changes in our revised manuscript (Page 5, Lines 165-166)

Comments 9: Line 388 – lack of the word 'FITC'.

Response 9: As you pointed out, we have added the word ‘FITC’ in Page 11, Line 417 of our revised manuscript.

  1. Response to Comments on the Quality of English Language

Point 1: The language needs to be checked and revised thoroughly.

Response 1: As you suggested, we have asked help from a native English-speaker with some knowledge in Poultry reproduction and physiology to improve the language of our manuscript. Please see the detailed changes in our revised manuscript.

Reviewer 2 Report

Comments and Suggestions for Authors

Positive notes to the authors:

1. The topic of the manuscript is current, as it is related to a study of the Effects of Recombinant and Native Prolactin on the Proliferation and Apoptosis of Goose Granulosa Cells;

2. The level of molecular biology, cytology and biochemistry has been tracked up to now in the field of avian proliferation and apoptosis, because prolactin plays a key role in regulation of incubation behavior, hormone secretion, and reproductive activities;

3. The purpose of the research is clearly formulated, which makes the manuscript completely completed according to the requirements of the scientific journal;

4. In the Material and methods section, the manipulations and the performed analyzes are described in detail;

5. The results are presented in a very well-designed section;

6. A relatively full-fledged discussion was held in which previous own and foreign researches were compared;

Negative notes and recommendations to the authors:

1. Despite the well-designed Results section, some figures are not formatted well. I recommend a 15-20% zoom in Figures 2, 3, 4, 5, 6 and 7 for better visualization;

2. The erudition of the authors is evident in the kasatta discussion. In this aspect, they could dig deeper into the bowels of molecular biology, genetics and biochemistry to better explain the effects of prolactin;

3. The conclusions drawn show the usefulness of this scientific research, but no concrete recommendations have been formulated for the poultry industry;

4. I recommend further molecular biological studies to further elucidate some reproductive mechanisms and to better understand the roles of PRL during avian ovarian follicle development.

Comments on the Quality of English Language

The article is written in relatively good English. However, I recommend finalizing and polishing by an English-speaking editor.

Author Response

Response to Reviewer 2 Comments

  1. Summary

We thank you for your careful review of our paper (Manuscript Number: IJMS-2683907) entitled “Comparison of the effects of recombinant and native prolactin on the proliferation and apoptosis of goose granulosa cells”. According to your comments, we have made corresponding responses on a ‘point by point’ basis that are listed as follows, and the detailed revisions have also been red highlighted in this revised manuscript. We hope that this revised manuscript will now meet the publication standard.

Thank you for your careful review and valuable comments again!

Sincerely,

Donghang Deng

2.  Questions for General Evaluation

Reviewer’s Evaluation

Does the introduction provide sufficient background and include all relevant references?

Yes

Are all the cited references relevant to the research?

Yes

Is the research design appropriate?

Can be improved

Are the methods adequately described?

Yes

Are the results clearly presented?

Can be improved

Are the conclusions supported by the results?

Can be improved

  1. Response to Comments and Suggestions

Positive notes to the authors:

  1. The topic of the manuscript is current, as it is related to a study of the Effects of Recombinant and Native Prolactin on the Proliferation and Apoptosis of Goose Granulosa Cells;
  2. The level of molecular biology, cytology and biochemistry has been tracked up to now in the field of avian proliferation and apoptosis, because prolactin plays a key role in regulation of incubation behavior, hormone secretion, and reproductive activities;
  3. The purpose of the research is clearly formulated, which makes the manuscript completely completed according to the requirements of the scientific journal;
  4. In the Material and methods section, the manipulations and the performed analyzes are described in detail;
  5. The results are presented in a very well-designed section;
  6. A relatively full-fledged discussion was held in which previous own and foreign researches were compared;

Response 1: We thank you very much for your valuable comments and approval of this article.

Negative notes and recommendations to the authors:

Comments 1: Despite the well-designed Results section, some figures are not formatted well. I recommend a 15-20% zoom in Figures 2, 3, 4, 5, 6 and 7 for better visualization;

Response 1: As you recommended, we have carefully re-formatted the figures 2-7 for better visualization. Please see the corresponding changes in our revised manuscript.

Comments 2: The erudition of the authors is evident in the kasatta discussion. In this aspect, they could dig deeper into the bowels of molecular biology, genetics and biochemistry to better explain the effects of prolactin;

Response 2: As you pointed out, we have appropriately expanded and improved the corresponding part in the discussion section. Please see the detailed changes in Page 8, Lines 238-254 of our revised manuscript.

Comments 3: The conclusions drawn show the usefulness of this scientific research, but no concrete recommendations have been formulated for the poultry industry;

Response 3: As you pointed out, we have improved the Conclusions section by supplementing both the theoretical and practical values of the present study. Please see the detailed changes in Page 13, Line 451-453 of our revised manuscript.

Comments 4: I recommend further molecular biological studies to further elucidate some reproductive mechanisms and to better understand the roles of PRL during avian ovarian follicle development.

Response 4: We firstly thank you for your valuable suggestions. Based on the highly-purified goose endogenously native prolactin obtained in the present study, we are currently striving to further isolate and purify the glycosylated isoform from goose native prolactin, as our previously published articles have shown that glycosylation influences the bioactivity and function of poultry prolactin. Furthermore, we are planning to systematically investigate the mechanisms by which prolactin and its glycosylation regulate goose ovarian follicular cell functions using the recombinant, native, and glycosylated prolactin simultaneously in our follow-up experiments. Hence, this study was mainly focused on the isolation and purification of goose recombinant and native prolactin as well as their effects on goose ovarian granulosa cell proliferation and apoptosis.

  1. Response to Comments on the Quality of English Language

Point 1: The article is written in relatively good English. However, I recommend finalizing and polishing by an English-speaking editor.

Response 1: As you suggested, we have asked help from a native English-speaker with some knowledge in Poultry reproduction and physiology to improve the language of our manuscript. Please see the detailed changes in our revised manuscript.

Reviewer 3 Report

Comments and Suggestions for Authors

The manuscript is an interesting read. The authors prominently isolated and purified nPRL from goose’ pituitary, compared nPRL and rPRL, and investigated their effects on GCs by testing cell viability assay and apoptosis at different concentrations. Comments:

>> Hypothalamic GnRH regulates gonadotropins (LH & FSH) which control sex steroids biosynthesis in the ovary. It has been suggested PRL controls steroid hormones. In context to goose nPRL, how can it be associated with impact of seasonal changes?  

>> Dose selection criteria?? The consideration of doses and its explanation is weak. The authors referred to insulin sensitivity, endometrial growth, and U266Myeloma cells concerning doses, however the same doses may not have been used in these studies as the authors used in the present study. At least one aspect of it should be validated to reach a conclusion.

>> Why was only Becl1 considered for investigating autophagy process? Does its reduction alone promote apoptosis? How? Many autophagy biomarkers like ATG-5, ATG12, p62, Lc3-II, LAMP1 etc. should have been tested for autophagy inhibition. Also, would you explain the nexus of autophagy and apoptosis with findings in the present study?

>> Some grammatical errors should be corrected in the manuscript.

Comments on the Quality of English Language

The English language may be improved. Grammatical errors should be corrected.

Author Response

Response to Reviewer 3 Comments

  1. Summary

We thank you for your careful review of our paper (Manuscript Number: IJMS-2683907) entitled “Comparison of the effects of recombinant and native prolactin on the proliferation and apoptosis of goose granulosa cells”. According to your comments, we have made corresponding responses on a ‘point by point’ basis that are listed as follows, and the detailed revisions have also been red highlighted in this revised manuscript. We hope that this revised manuscript will now meet the publication standard.

Thank you for your careful review and valuable comments again!

Sincerely,

Donghang Deng

2.  Questions for General Evaluation

Reviewer’s Evaluation

Does the introduction provide sufficient background and include all relevant references?

Can be improved

Are all the cited references relevant to the research?

Yes

Is the research design appropriate?

Can be improved

Are the methods adequately described?

Yes

Are the results clearly presented?

Yes

Are the conclusions supported by the results?

Can be improved

  1. Response to Comments and Suggestions

Comments 1: The manuscript is an interesting read. The authors prominently isolated and purified nPRL from goose’ pituitary, compared nPRL and rPRL, and investigated their effects on GCs by testing cell viability assay and apoptosis at different concentrations.

Response 1: We thank you very much for your valuable comments and approval of this article.

Comments 2: Hypothalamic GnRH regulates gonadotropins (LH & FSH) which control sex steroids biosynthesis in the ovary. It has been suggested PRL controls steroid hormones. In context to goose nPRL, how can it be associated with impact of seasonal changes?

Response 2: We are in full agreement with you that hypothalamic GnRH regulates gonadotropins (LH & FSH) which control sex steroids biosynthesis in the ovary, and PRL has been suggested to control steroid hormones. In context to goose endogenously native PRL (nPRL), which is mainly synthesized and secreted by lactotroph cells in the anterior pituitary gland, it has been well documented that seasonal and photoperiodic regulation of secretion of nPRL is strongly associated with the ovarian activities and egg production performance in a number of indigenous domestic goose breeds [1-6]. In this respect, it is generally believed that the reproductive seasonality of domestic geese is closely related to the pituitary secretions of both gonadotrophins and nPRL in response to photoperiodic changes. Also, we have supplemented such information in the second and third paragraphs of the Introduction section of our revise manuscript.

References

  1. Shi, Z.; Tian, Y.; Wu, W.; Wang, Z., Controlling reproductive seasonality in the geese: a review. World's Poultry Science Journal 2008, 64, (3), 343-355.
  2. Huang, Y. M.; Shi, Z. D.; Liu, Z.; Liu, Y.; Li, X. W., Endocrine regulations of reproductive seasonality, follicular development and incubation in Magang geese. Animal Reproduction Science 2008, 104, (2-4), 344-358.
  3. Yang, H. M.; Wang, Y.; Wang, Z. Y.; Wang, X. X., Seasonal and photoperiodic regulation of reproductive hormones and related genes in Yangzhou geese. Poultry Science 2016, 486.
  4. Yao, Y.; Yang, Y. Z.; Gu, T. T.; Cao, Z. F.; Chen, G. H., Comparison of the broody behavior characteristics of different breeds of geese. Poultry Science 2019, 98, (11), 5226-5233.
  5. Chen, R.; Guo, R.; Zhu, H.; Shi, Z., Development of a sandwich ELISA for determining plasma prolactin concentration in domestic birds. Domestic animal endocrinology 2019, 67, 21-27.
  6. Zhao, W.; Yuan, T.; Fu, Y.; Niu, D.; Lu, L., Seasonal differences in the transcriptome profile of the Zhedong white goose (Anser cygnoides) pituitary gland. Poultry Science 2020.

Comments 3: Dose selection criteria?? The consideration of doses and its explanation is weak. The authors referred to insulin sensitivity, endometrial growth, and U266Myeloma cells concerning doses, however the same doses may not have been used in these studies as the authors used in the present study. At least one aspect of it should be validated to reach a conclusion.

Response 3: As you pointed out, we have supplemented such information in the cell culture and treatment part of the Materials and Methods section. Please see the detailed changes in our revised manuscript (Page 11, Lines 392-395).

Comments 4: Why was only Becl1 considered for investigating autophagy process? Does its reduction alone promote apoptosis? How? Many autophagy biomarkers like ATG-5, ATG12, p62, Lc3-II, LAMP1 etc. should have been tested for autophagy inhibition. Also, would you explain the nexus of autophagy and apoptosis with findings in the present study?

Response 4: As you pointed out, Becn1, known as an autophagy-related protein, which binds to Bcl-2, results in preventing assembly of the pre-autophagosomal structure and inhibition of autophagy[1]. It has been reported that autophagy is involved in regulating the apoptosis of GCs to accelerate follicular atresia [2]. In this study, the expression levels of Becn1 and Bcl-2 in phGCs and hGCs significantly decreased after treatment with high concentrations of either rPRL or nPRL. There is evidence that Becn1 plays important roles in the regulation of the life span of human corpus luteum and ovarian androgen-secreting cells by maintaining autophagy at levels promoting cell survival rather than cell death [3]. Additionally, the siRNA-mediated knockdown of Becn1 showed that inhibition of autophagy resulted in decreased expression of genes associated with GCs differentiation, as well as in reduced estradiol synthesis[4]. These results implied that high PRL concentrations may promote the apoptosis of goose by disrupting the process of autophagy in GCs. Also, we have supplemented such information in Pages 9, Lines 287-299 of our revised manuscript.

References

  1. Marquez, R. T.; Xu, L., Bcl-2:Beclin 1 complex: multiple, mechanisms regulating autophagy/apoptosis toggle switch. American journal of cancer research 2012, 2, (2), 214.
  2. Choi, J.; Jo, M.; Lee, E.; Choi, D., AKT is involved in granulosa cell autophagy regulation via mTOR signaling during rat follicular development and atresia. Reproduction 2014, 147, (1), 73-80.
  3. Gaytán, M.; Morales, C.; Sánchez-Criado, J.; Gaytán, F., Immunolocalization of beclin 1, a bcl-2-binding, autophagy-related protein, in the human ovary: possible relation to life span of corpus luteum. Cell tissue research 2008, 331, (2), 509-17.
  4. Shao, T.; Ke, H.; Liu, R.; Xu, L.; Han, S.; Zhang, X.; Dang, Y.; Jiao, X.; Li, W.; Chen, Z.; Qin, Y.; Zhao, S., Autophagy regulates differentiation of ovarian granulosa cells through degradation of WT1. Autophagy 2022, 18, (8), 1864-1878.

Comments 5: Some grammatical errors should be corrected in the manuscript.

Response 5: As you pointed out, we have asked help from a native English-speaker with some knowledge in Poultry reproduction and physiology to correct the grammatic errors and polish the language of our manuscript. Please see the detailed changes in our revised manuscript.

  1. Response to Comments on the Quality of English Language

Point 1: The English language may be improved. Grammatical errors should be corrected.

Response 1: As you suggested, we have asked help from a native English-speaker with some knowledge in Poultry reproduction and physiology to finalize and polish the language of our manuscript. Please see the detailed changes in our revised manuscript.

Round 2

Reviewer 1 Report

Comments and Suggestions for Authors

All my comments/suggestions were taken into consideration by the authors of the paper. I have no further comments.